# Cancer Screening among Rural People Who Use Drugs: Colliding Risks and Barriers

**DOI:** 10.3390/ijerph19084555

**Published:** 2022-04-10

**Authors:** Wiley D. Jenkins, Jennifer Rose, Yamile Molina, Minjee Lee, Rebecca Bolinski, Georgia Luckey, Brent Van Ham

**Affiliations:** 1Department of Population Science and Policy, Southern Illinois University School of Medicine, Springfield, IL 62794, USA; mlee88@siumed.edu; 2Department of Family and Community Medicine, Southern Illinois University School of Medicine, Carbondale, IL 62901, USA; jrose@siumed.edu (J.R.); gluckey34@siumed.edu (G.L.); 3Division of Community Health Sciences, University of Illinois at Chicago School of Public Health, Chicago, IL 60612, USA; ymolin2@uic.edu; 4Simons Cancer Institute, Southern Illinois University School of Medicine, Springfield, IL 62702, USA; 5Department of Sociology, Southern Illinois University, Carbondale, IL 62901, USA; rebecca81110@siu.edu; 6Center for Rural Health and Social Services Development, Southern Illinois University, Carbondale, IL 62901, USA; bvanham49@siumed.edu

**Keywords:** rural cancer disparities, rural cancer screening, cancer screening and adherence

## Abstract

Rural cancer disparities are associated with lesser healthcare access and screening adherence. The opioid epidemic may increase disparities as people who use drugs (PWUD) frequently experience healthcare-associated stigmatizing experiences which discourage seeking routine care. Rural PWUD were recruited to complete surveys and interviews exploring cancer (cervical, breast, colorectal, lung) risk, screening history, and healthcare experiences. From July 2020–July 2021 we collected 37 surveys and 8 interviews. Participants were 24.3% male, 86.5% White race, and had a mean age of 44.8 years. Females were less likely to report seeing a primary care provider on a regular basis, and more likely to report stigmatizing healthcare experiences. A majority of females reporting receiving recommendations and screens for cervical and breast cancer, but only a minority were adherent. Similarly, only a minority of males and females reported receiving screening tests for colorectal and lung cancer. Screening rates for all cancers were substantially below those for the US generally and rural areas specifically. Interviews confirmed stigmatizing healthcare experiences and suggested screening barriers and possible solutions. The opioid epidemic involves millions of individuals and is disproportionately experienced in rural communities. To avoid exacerbating existing rural cancer disparities, methods to engage PWUD in cancer screening need to be developed.

## 1. Introduction

Approximately 19% of the US population (~60 million people) resides in rural areas, and faces significant disparities regarding cancer screening, incidence, and mortality [1]. Recent studies report that rural individuals experience increased incidence rates, and later stages of diagnosis, of cancers of the colon and rectum (CRC), breast (BC), and cervix (CC) [2,3,4]. During 2006–2015, the annual age-adjusted death rates for all cancer sites combined decreased at a slower pace in nonmetropolitan areas (−1.0% per year) than in metropolitan areas (−1.6% per year), increasing the differences in these rates [5].These cancers are also associated with modifiable risk factors and behaviors (e.g., tobacco and alcohol use, human papillomavirus (HPV) infection, obesity, screening utilization), many of which more frequently occur in rural communities [3,6,7,8]. Cumulatively, rural disparities in cancer incidence and mortality are significant and increasing.

Compounding modifiable risks, rural residents also face a multitude of barriers to healthcare access, including cultural norms, financial constraints, limited services, and insufficient public transportation [9,10,11,12,13,14,15,16,17]. Physician shortages and Health Profession Shortage Areas are more common, with only 12% of primary care physicians working in rural areas (and 8% of specialty physicians) [18]. Further, while the overall number of physicians in the US grew by 16,000 during 2013–2015, the rural total dropped by 1400, and 26% of rural residents report not getting needed healthcare in the past few years [18,19]. Physical access is also decreasing, and since 2010 nearly 7% (120+) of rural hospitals have closed and another 25% face closure (this report before the COVID-19 pandemic and associated financial impact on healthcare organizations) [20]. These shortages likely contribute to findings that rural residents are less likely to report recommendations for, or receipt of, preventive services and cancer screening [11,21]. In spite of increased cancer risk and incidence in rural areas, access to healthcare is increasingly difficult to obtain.

Another contemporary epidemic is that associated with opioid and other illicit drug use. The extent of this epidemic is indicated in various reports showing that: 23.9% of those aged 18–25 reported using illicit drugs during the past month (2018); ~23 million US residents experienced substance use disorder (SUD) at some point (2015); and 2.6% of the US population aged 13 years or older had ever injected drugs and clinical studies indicate this may be increasing [22,23,24,25]. Ultimately, much of drug use and related adverse health outcomes are disproportionately experienced in rural areas, which face increased rates of use of opioids, alcohol, cigarettes, smokeless tobacco, and methamphetamine [26,27]. Further, clinicians often have high levels of stigma and negative attitudes towards patients with substance use disorders, which lessen both treatment outcomes and patient empowerment [28,29]. Subsequently reflected in the patient experience (via healthcare interactions and other situations), internalized stigma has been found to have negative impacts on individual mental health and health seeking behavior [30,31,32,33,34,35,36,37,38,39,40]. The US drug use epidemic is impacting millions of residents, disproportionately so in rural areas with already-diminished healthcare access. Further, widely prevalent healthcare-associated stigma towards substance use, and the internalization of such stigma, serves as both a barrier to proper treatment receipt and even treatment seeking among people who use drugs (PWUD). Members of our team have documented high levels of PWUD stigma among rural communities, which may be exacerbated during the COVID-19 pandemic [41,42].

We argue that rural PWUD face at least equal cancer risk as other rural individuals but may also be less likely to receive recommended screenings due in part to stigmatizing healthcare experiences. The purpose of this mixed-methods study was to explore how demographics, stigmatizing healthcare experiences, and cancer experience are associated with cancer screening recommendations, screening receipt, and current screening adherence (i.e., up-to-date) among rural people who use drugs.

## 2. Materials and Methods

### 2.1. Recruitment

For this initial exploration of cancer risk and screening knowledge we utilized a mixed-methods approach whereby a convenience sample of individuals was recruited to complete a survey, and a subset of these same participants subsequently invited to participate in an interview. Participants were recruited from the existing *Ending Transmission of HIV, HCV, and STDs and Overdose in Rural Communities of People who Inject Drugs* (ETHIC) clinical trial (NCT04427202) from July 2020 through July 2021. ETHIC eligibility includes: age ≥ 15 years; injection (any drug) OR non-injection opioid use to get high within the past 30 days; English-speaking; and ability to provide informed consent. Additional criteria for inclusion in this study included being eligible for at least one cancer screen due to age and/or smoking history. The age ranges were determined according to the United States Preventive Services Task Force screening recommendations, which are: females only aged 21–65 years (cervical) and 50–74 (breast); and males and females aged 50–75 years (colorectal) and 50–80 years plus 20 pack-year smoking history (lung) [43,44,45,46].

### 2.2. Surveys

Survey sections explored participant demographics; cancer screening knowledge and ever receipt of clinician recommendation for screening; screening history and barriers; and personal and family cancer history. We also asked four questions specifically related to healthcare access and use in relation to their drug use. Termed stigmatizing healthcare experiences (SHE), they were as follows: SHE-1: Have you ever avoided health care because of your drug use?; SHE-2: Have you ever had a healthcare worker refuse to treat you or denied you access to medical treatment or care because of your drug use?; SHE-3: Do you feel ‘accepted’ and ‘non-judged’ during office visits?; and SHE-4: Have you ever had bad experiences with primary care such that you considered not going anymore?

### 2.3. Interviews

Interviews were designed to elicit more contextual aspects of cancer risk and screening access and utilization. Interviews lasted about fifteen minutes and were audio-recorded by placing a digital recorder next to the telephone, Interviews were conducted (and previous surveys as well) by study staff experienced with both as part of the larger ETHIC clinical trial from which these participants were recruited. The interview guide and survey instrument are available online at Appendix A. 

### 2.4. Data Analysis

Survey data were subjected to descriptive, *t*-test, and cross tabulation analysis in SPSS. Due to the limited number of surveys and wide variation in some data, more complex analyses such as regression and modeling were not justified. Interview recordings were supplemented with detailed written notes. These field notes were subsequently verified and expanded by listening back to interview recordings, coded, and subjected to preliminary thematic analysis.

Due to COVID-19 social distancing requirements, we were unable to perform in-person recruitment or meet individuals during normal service delivery. Thus, recruitment for the survey was substantially hindered, and consisted entirely of calling and texting current ETHIC participants. We then selected a convenience sample of survey participants to invite for an interview, with selection seeking to maximize diversity in gender, race, and drug of choice. All surveys and interviews were conducted by phone and data entered into REDCap. Participants were reimbursed for their time ($20 for the survey, and an additional $20 if selected for interview). This study was approved by the University of Chicago Institutional Review Board.

## 3. Results

From 27 July 2020 to 13 July 2021 we invited 37 current ETHIC enrollees to participate in the survey, and all (100%) agreed. Participants were 24.3% male, 86.5% White race, and had a mean age of 44.8 years (range of 23–66; Table 1). Due to gender-based differences in cancer screening eligibility, male participants were aged 51–66 years while females were aged 23–62 years. Distribution of race, marital status, and education level did not significantly differ between males and females, though age was significantly different (*p* < 0.001; as expected). All but one participant reported an income <$25,000. Regarding behaviors, the majority reported: no past month alcohol consumption; moderate physical activity 4.7 days/week; and past month participation in physical activity (no significant differences). A large majority also reported past and/or current smoking, with a significant difference in average pack-years (*p* = 0.016; the difference is nonsignificant if comparison limited to those aged 50+ years). Regarding healthcare engagement and experiences, differences in engagement in routine care, care avoidance due to drug use, and feelings of acceptance in the healthcare environment were observed but not statistically significant. Finally, there are no significant differences in reported personal cancer diagnosis, or diagnosis among close family members.

We next explored receipt of cancer screening recommendations; if the individual ever received specific screenings; and if the individual was past due for screening due to age, risk, and length of time since last reported screen (Table 2).

### 3.1. Cervical Cancer Screening

All 28 female participants were eligible. The majority reported receiving a cervical cytology recommendation and also receiving the test. For the high risk HPV test, only a minority reported receiving either a recommendation or the test. A majority (53.6%) of females were past due for a cervical cancer screen.

### 3.2. Breast Cancer Screening

Seven of the female participants were eligible. The majority reported receiving a mammography recommendation and also receiving the test. Only one reported a recommendation for the breast MRI exam, and none reported receipt. A majority (57.1%) of females were past due for a breast cancer screen.

### 3.3. Colorectal Cancer Screening

Seven of the female and all the male participants (combined *n* = 16) were eligible. Approximately half of females and males received a recommendation for colonoscopy, and a minority of individuals reported recommendations for CT colonoscopy, sigmoidoscopy, and stool test. Less than half of individuals (6) reported ever receipt of a colonoscopy, small minorities reported receipt of CT colonoscopy (1) or stool (2) and none for sigmoidoscopy. A majority of both females and males (combined 80.0%) were past due for a colorectal cancer screen.

### 3.4. Lung Cancer Screening

Six of the female and 8 of the male participants (combined *n* = 14) were eligible. Only 1 female (and no males) had received a recommendation and reported screening receipt. All 16 (100%) were past due for a lung cancer screen.

From the survey participants we successfully recruited 8 individuals to complete interviews. Interviewees were 50% male, 87.5% White race, and reported a diversity of drug of choice (cocaine, opioids, methamphetamine; Table 3). Survey findings are consistent with data from participant interviews which illustrate a lack of familiarity with cancer screening guidelines, some ambivalence about the receipt of cancer screening recommendations, and difficulties with engaging in routine care.

Laura, a 50-year-old woman who injects methamphetamine, related her lack of knowledge around screening guidelines and being overdue for recommended screenings to her drug use saying, “*…in this area, if you are an addict, they—the percentage of people that actually go to the doctor is really low. We end up going to the community health clinics here, and they don’t, uh, tell you about the availability of these [cancer screenings] unless you already know them.*” Laura went on to say, “*…the stigma here at family practice, at [local hospital], is if you use methamphetamine, you are not gonna get treatment that is of quality, really…*”

Similarly, Billy, a 66-year-old man who endorses methamphetamine injection reported being “*shut out*” of the primary care office in his town due to his drug use history.

When asked about receiving recommended cancer screenings, Frank, a 58-year-old Black man who uses cocaine and opioids, stated, “*A lot of times, it’s kinda hard because, I had to go sixty-some miles to get that help, you know what I’m sayin’*?”

Taken together, interview participants questioned the quality of care they received due to their drug use, and some reported traveling hours, even out of state, to access healthcare and receive cancer screenings at hospitals where they felt they were treated more fairly.

Lastly, survey data were examined for associations between the two cancer history experiences (i.e., participant and family member past cancer diagnosis), the four stigmatizing healthcare experiences questions (SHE-1-4), and current screening status. While only one association was statistically significant (Table 4), the following points were noted.

Responses for SHE-1, SHE-2, and the cancer experiences are not consistent in indicating they may be a barrier or facilitation to screening.

Feeling accepted by their provider who knows about their drug use (SHE-3) is consistently more frequently reported by those current with CC, BC and CRC and less so by those past due.

Bad experiences in healthcare (SHE-4) are more frequently reported among those past due for CC, BC, and CRC.

Females who reported a family member with a cancer diagnosis were more likely to be up-to-date with cervical cancer screening, but no other cancer diagnosis (personal or family) was associated with any other cancer screen.

Interview participants expanded on their experiences with healthcare in their local areas, sharing narratives of both positive and negative encounters. Positive clinical interactions were more common among participants who were receiving medication for opioid use disorder from their primary care provider or those who travel outside of their local area to receive care. Still, many participants reported stigmatizing experiences in the healthcare setting.

When discussing a recent encounter in her local emergency department, Laura said, *“…the staff treats you like shit and then they laugh at you. You can actually hear them…they were reluctant to treat me, they wouldn’t listen to me, there was only one nurse there that actually treated me like I was a human being. They just were rushin’ me out the door*.”

Carol, a 62-year-old woman who uses opioids nonmedically echoed Laura’s sentiment, stating “*To me, if you don’t like people, then you shouldn’t be working with healthcare, you know?...They [hospital staff] were all very rude and opinionated about the kind of medicine you’ve already been maybe taking for years*.”

In this way, prior negative experiences in the clinical setting due to drug use status *and* the expectation of being treated poorly in the future created a barrier to engagement with healthcare that interrupted awareness of screening guidelines and receipt of screening tests.

## 4. Discussion

For this pilot study, conducted during the height of the COVID-19 pandemic, were able to recruit 37 rural people who use drugs to complete telephone administered surveys, and 8 then were selected for follow up interviews. Due to age differences in gender-based screening guidelines, there were differences in age and smoking pack years between males and females. Further, the majority of males (55.5%) were married or cohabitating versus a minority of females (34.2%). The data indicate some healthy and risk reduction behaviors, such as a majority reporting past month exercise and only a minority reporting past month alcohol use. There may be differences in stigmatizing healthcare experiences, but the significance is difficult to ascertain due to the small sample size. Still, a larger proportion of females report such experiences.

Receipt of screening recommendations and tests varied considerably by type. The majority of females reported receiving recommendations for cervical and breast cancer screening, and a majority also reported ever-receipt. Only approximately half of males and females reported receipt of colorectal cancer screening recommendations, and less than half reported ever receipt; and only one female and no males reported receipt of a lung cancer screening recommendation or test. While many reported ever-receipt of the various screens for which they were eligible, the majority (62.2%) were also past due for at least one screen (i.e., 23 individuals were past due for a total of 44 screens). In comparison, we observe that participant self-reported screening adherence for every screen is far less than that for Illinois, the United States, and rural areas in general (Table 2). Our data also indicated, but not to a statistically significant degree, that past stigmatizing experiences may be associated with screening nonadherence, and these indications from the survey data are supported by the interviews.

Interview data point to an important barrier to receipt of cancer screenings—transportation. Most participants report relying on friends or family for transportation to places they need to go. These arrangements are not always reliable, and many participants reported being “*blown off*”, resulting in them missing their appointments. Additionally, public transit, while usually reliable, was reported as problematic due to waiting times and now COVID-19. The data also offer some possible insight into modes of healthcare delivery that may be more acceptable for PWUD. When asked about the feasibility and acceptability of receiving healthcare, including some cancer screening tests, at a mobile or off-clinic site, nearly all participants vocalized that they would feel more comfortable and better able to access healthcare if offered through this venue.

In toto, the survey and interview data support the idea that rural people who use drugs not only experience the rural circumstance of less frequent cancer screening, but the compounding influence their drug use may have on healthcare seeking and receipt. Past stigmatizing experiences increase reticence to seek care, lessening an already-lower likelihood of receiving appropriate screening recommendations. The data show, for example, that only 39% of females report having a primary care provider they see on a regular or routine basis, and 57% report having past bad experiences with primary care such that they considered not going anymore. In this light, data indicating the majority of females (and males) are past due for every screen for which they are eligible, and interview comments suggestive that care provided in other venues might be preferred, are consistent and point to areas of further work and development. As the opioid and other drug use (e.g., methamphetamine) epidemic continues to grow and evolve, developing better means to satisfactorily engage people who use drugs in healthcare, and exploring alternative means whereby care can be provided, becomes increasingly critical. The scale of those at compounding risk for missed/delayed screening is substantial, with estimates of past year use among those aged 12+ years of methamphetamine, heroin, and prescription pain killers (misuse) at 2.5 million, 90,000, and 9.3 million, respectively [55]. Further, there are some studies indicating that drug use may be associated with increased cancer risk [56,57,58]. As already shown, rural areas are disproportionately impacted by drug use, and so to avoid increasing the existing rural disparities in cancer screening new means of engagement and utilization, specifically developed for the rural environment should be explored [26,27].

There are limitations to this work. The survey data are too few for more rigorous statistical analyses (e.g., insufficient variation in sexual orientation; drug of choice; frequency and methods of use). More extensive data would allow for exploration regarding how screening utilization might vary by important variables such as type(s) of drug use, age and previous experiences, and other stigmatizing characteristics; and how negative experiences/perceptions might be countered by intervention. From the interviews, the data are not sufficient to reach saturation, and findings are preliminary. Future interviews should explore these emergent themes more fully, particularly in relation to cancer screening priority, as preliminary findings indicate some indifference toward cancer screening adherence.

## 5. Conclusions

In conclusion, this work may be among the first to explicitly examine cancer screening experiences among rural people who use drugs. The drug use epidemic in the US is continuing to grow, often disproportionately impacting rural communities, and exacerbating existing disparities in healthcare utilization and cancer screening utilization. Our preliminary data indicates that rural people who use drugs may need specific consideration for interventions to increase cancer screening uptake, as the stigmatizing nature of their drug use adds additional barriers to seeking already-diminishing care in rural areas.

## Figures and Tables

**Table 1 ijerph-19-04555-t001:** Participant demographics, alcohol and tobacco use, and experiences with healthcare and cancer (*n* = 37).

Variable					Male (*n* = 9)	Female (*n* = 28)	*p*
Demographics	Age (mean; range) **				57.1; 51–66	40.8; 23–62	<0.001
	Race (% White race)				88.9%	85.7%	0.648
	Marital status **			Never married	0	21.4	0.346
				Married	22.2	17.9	
				Separated/divorced/widowed	44.4	46.4	
				cohabitating	33.3	14.3	
	Education level **			Elementary	11.1	0	0.085
				HS	33.3	64.3	
				college	55.6	35.7	
Behaviors	During the past 30 days, how many days per week did you have at least one drink of any alcoholic beverage? (% none)				66.7	64.3	0.614
	In a typical week, how many days do you do any physical activity or exercise of at least moderate intensity? (mean)				4.7	4.7	0.965
	During the past month, did you participate in any physical activities or exercises such as running, yoga, golf, gardening, or walking for exercise? (% yes)				77.8	85.7	0.457
	Do you now, or have you in the past, smoked cigarettes? (% yes)				88.9	96.4	0.432
	Average pack years **				33.6	22.9	0.016 ^§^
Healthcare engagement	Do you currently see a primary care provider on a regular or routine basis? (% yes)				66.7	39.3	0.147
	How long have you been going to this provider?		Less than 6 months		0.0	9.1	0.623
			At least 6 months but less than 1 year		16.7	0.0	
			At least 1 year but less than 3 years		16.7	27.3	
			At least 3 years but less than 5 years		33.3	27.3	
			5 years or more		33.3	36.4	
Stigmatizing healthcare experiences (SHE)	SHE-1: Have you ever avoided health care because of your drug use? (% yes)				44.4	75.0	0.100
	SHE-2: Have you ever had a healthcare worker refuse to treat you or denied you access to medical treatment or care because of your drug use? (% yes)				22.2	35.7	0.376
	SHE-3: Do you feel ‘accepted’ and ‘non-judged’ during office visits?	No, they treat me differently			22.2	46.4	0.208
		Yes, and they know about the drug use			44.4	42.9	
		Yes, but they don’t know about the drug use			33.3	10.7	
	SHE-4: Have you ever had bad experiences with primary care such that you considered not going anymore? (% yes)				44.4	57.1	0.388
Cancer experiences	Have you ever been told by a health care provider that you have a cancer? (% yes)				11.1	17.9	0.543
	Has a parent, sibling, or child related to you by blood ever been diagnosed with cancer? (Include only siblings with same biological mother and father). (% yes)				66.7	67.9	0.829

** items are expected to differ due to different age ranges of screening eligibility for males and females. ^§^ This difference is insignificant if only assessed among those aged 50+ years.

**Table 2 ijerph-19-04555-t002:** Reported receipt of screening recommendation, specific cancer screening, and current screening status (adherent versus past due; with comparison to IL and US adherent rates).

Gender	Cancer Site (Eligibility; # Eligible)	Recommendation? (*n*; %)	Ever Receipt? *n* (%)	Current Screening Adherence (%)			
				Participants	IL	US (All)	US (Rural vs. Urban)
Females	Cervical (age 21–65; *n* = 28)	Cervical cytology = 24 (85.7)	23 (82.1)	46.4	79.4 [47]	80.2 [48] 81.3 [47]	77.7 vs. 84.4 [49]
		HPV DNA = 6 (21.4)	5 (17.9)				
	Breast (age 50–74; *n* = 7)	Mammography = 6 (88.7)	5 (71.4)	42.9	78.1 [47]	78.3 [48] 71.7 [47]	81 vs. 81 [50]
		MRI = 1 (14.3)	0 (0)				
	Colorectal (age 50–75; *n* = 7)	Colonoscopy = 3 (42.9)	3 (42.9)	14.3	75.9 [51]	73.5 [52] 63.4 [47]	78 vs. 82 [50] 65.5 vs. 68.2 [51] ^§^
		CT = 0 (0)	0 (0)				
		Sigmoidoscopy = 0 (0)	0 (0)				
		Stool = 1 (14.3)	0 (0)				
	Lung (age 50–80 + 20 pack-year; *n* = 6)	LDCT = 1 (16.7)	1 (16.7)	0	6.3 [53] ^§^	5.7 [54] ^§^ 17.5 [53] ^§^	16.3 vs. 17.7 [53] ^§^
Males	Colorectal (age 50–75; *n* = 9)	Colonoscopy = 5 (55.6)	3 (33.3)	33.3	62.0 [51]	71.2 [52] 61.9 [47]	65.5 vs. 68.2 [51] ^§^
		CT = 2 (22.2)	1 (11.1)				
		Sigmoidoscopy = 0 (0)	0 (0)				
		Stool = 4 (44.4)	2 (22.2)				
	Lung (age 50–80 + 20 pack-year; *n* = 8)	LDCT = 0 (0)	0 (0)	0	6.3 [53] ^§^	5.7 [54] ^§^ 17.5 [53] ^§^	16.3 vs. 17.7 [53] ^§^

^§^ data are for combined sexes.

**Table 3 ijerph-19-04555-t003:** Selected interviewee characteristics.

Pseudonym	Age	Sex	Race	Drug of Choice
Billy	66	Male	White	methamphetamine
Calvin	59	Male	White	opioids
Carol	62	Female	White	opioids
Frank	58	Male	Black	cocaine
Harry	61	Male	White	cocaine
Julia	50	Female	White	methamphetamine
Laura	50	Female	White	methamphetamine
Sheryl	42	Female	White	opioids

**Table 4 ijerph-19-04555-t004:** Association between healthcare experiences and personal and family cancer history with screening adherence.

	Cervical			Breast			Colorectal			Lung		
	Past Due (*n* = 15)	Current (*n* = 13)	*p*	Past Due (*n* = 4)	Current (*n* = 3)	*p*	Past Due (*n* = 12)	Current (*n* = 4)	*p*	Past Due (*n* = 14)	Current (*n* = 0)	*p*
SHE-1: Have you ever avoided health care because of your drug use? (% yes)	80.0	69.2	0.412	75.0	100	0.571	58.3	75.0	0.511	57.1	N/A	N/A
SHE-2: Have you ever had a healthcare worker refuse to treat you or denied you access to medical treatment or care because of your drug use? (% yes)	33.3	38.5	0.544	75.0	33.3	0.371	41.7	25.0	0.511	42.9	N/A	N/A
SHE-3: Do you feel ‘accepted’ and ‘non-judged’ during office visits? (% “yes, and they know about the drug use” vs. - “yes, but they don’t know about the drug use”, or - “no, they treat me differently”)	26.7	61.5	0.171	25.0	33.3	0.084	25.0	75.0	0.155	28.6	N/A	N/A
SHE-4: Have you ever had bad experiences with primary care such that you considered not going anymore? (% yes)	66.7	46.2	0.239	100	66.7	0.429	66.7	50.0	0.489	71.4	N/A	N/A
Have you ever been told by a health care provider that you have a cancer? (% yes)	20.0	15.4	0.572	25.0	0	0.571	8.3	25.0	0.450	14.3	N/A	N/A
Has a parent, sibling, or child related to you by blood ever been diagnosed with cancer? (Include only siblings with same biological mother and father). (% yes)	53.3	91.7	0.038 *	50.0	100	0.286	83.3	25.0	0.063	64.3	N/A	N/A

* The single response of “I don’t know/am unsure” was removed here.

## Data Availability

The data presented in this study are available on request from the corresponding author. The data are not publicly available due to the sensitive nature of the data collected and other study aims exploring illicit drug use.

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
