# Peer review of "Cancer Screening among Rural People Who Use Drugs: Colliding Risks and Barriers"

_ijerph, 2022, doi:10.3390/ijerph19084555_

Round 1
Reviewer 1 Report
The study is interesting.
I have a few remarks.
1) Methods & Materials
It will be easier to follow if the method sections are spited, or separated in sub-headings e,g, method, what kind of methods used and why was the method chosen, with references, recruitment/ settings, Data analysis and ethical issues/ consideration.
Also, sentences in page 2 lines 98-99 is understandable ("Participants each received $20 for their time completing surveys and interviews").
Results: these sentences page 3, lines 115-116, " From 27 Jul 2020 to 13 Jul 2021 we were able to recruit 37 survey participants. All 115 those approached agreed to participate (100%), I think these belongs to methods section. Also, the whole sentences need to be reframed, it does not make sense.
Discussion: page 8, lines 26-27, The sentences “A majority of both sexes reported not using alcohol in the past month, and also past month exercise, does not make sense.
Author Response
It will be easier to follow if the method sections are spited, or separated in sub-headings e,g, method, what kind of methods used and why was the method chosen, with references, recruitment/ settings, Data analysis and ethical issues/ consideration.
- The Methods section has been revised for clarity.
Also, sentences in page 2 lines 98-99 is understandable ("Participants each received $20 for their time completing surveys and interviews").
- RESPONSE: This was re-worded to now state: Participants were reimbursed for their time ($20 for the survey, and an additional $20 if selected for interview)
Results: these sentences page 3, lines 115-116, " From 27 Jul 2020 to 13 Jul 2021 we were able to recruit 37 survey participants. All those approached agreed to participate (100%), I think these belongs to methods section. Also, the whole sentences need to be reframed, it does not make sense.
- This was re-worded to now state: From 27 Jul 2020 to 13 Jul 2021 we invited 37 current ETHIC enrollees to participate, and all (100%) agreed.
Discussion: page 8, lines 26-27, The sentences “A majority of both sexes reported not using alcohol in the past month, and also past month exercise, does not make sense.
- This was re-worded to now state: The data indicate some healthy and risk reduction behaviors, such as a majority reporting past month exercise and only a minority reporting past month alcohol use.
Reviewer 2 Report
Congratulations on an interesting and useful paper for future practice. I have some minor comments that could be addressed to improve the paper:
Introduction
Line 35 “Approximately 19% of the U.S. population (~60 million people) resides” should read “Approximately 19% of the U.S. population (~60 million people) reside”
Line 43 Spelling error: beahviors
Line 44 HPV should be defined
Need to be consistent in use of US and U.S.
Line 74 HC should be defined
Methods
Survey methods are relatively well described though I would recommend including copy of the survey, unless all the questions are covered in Table 2 (and open-ended interview questions) as appendices. Also unclear how many items were in the survey, who conducted the survey – were they trained, how was the survey designed or was it an existing instrument and, if so, has it been previously validated for use in this type of population?
How were people selected for interviews? This should be described – are they the same people who took the surveys? Were they purposively sampled.
Methods are also a little underdone for qualitative, who did the interviews (were they independent of the clinical trial), who did the analysis, was coding independently triangulated? Suggest using COREQ checklist to ensure data trustworthiness can be demonstrated.
Results:
I do not agree with findings being presented as bullet points and should be written in regular paragraph styles
Line 29 – what is SHE? Is this referring to survey items?
Be consistent with excerpts – all in italics
Otherwise well described
Discussion is well written and picks up on key findings in the context of existing literature.
Author Response
Line 35 “Approximately 19% of the U.S. population (~60 million people) resides” should read “Approximately 19% of the U.S. population (~60 million people) reside”
- I think for the purposes of verb conjugation, the items in parentheses is not considered. So, 19% of the US population resides… (versus people) reside) Will follow editorial preference on this.
Line 43 Spelling error: beahviors
Line 44 HPV should be defined
Need to be consistent in use of US and U.S.
Line 74 HC should be defined
- These have all been corrected.
Survey methods are relatively well described though I would recommend including copy of the survey, unless all the questions are covered in Table 2 (and open-ended interview questions) as appendices. Also unclear how many items were in the survey, who conducted the survey – were they trained, how was the survey designed or was it an existing instrument and, if so, has it been previously validated for use in this type of population?
- The survey instrument and interview guide have been submitted as appendices. The survey components are a combination of common demographic questions (e.g., age, race), healthcare engagement questions drawn from our experiences with how people who use drugs interact with healthcare, and common questions about cancer screening knowledge and experience. They are not necessarily drawn from a single validated instrument. The individuals conducting the surveys (and interviews) have been doing so with this population for several years as part of the larger/background ETHIC clinical trial.
How were people selected for interviews? This should be described – are they the same people who took the surveys? Were they purposively sampled.
- This convenience sample, selected to maximize diversity in gender, race, and drug of choice is now better described as follows in Results: We then selected a convenience sample of survey participants to invite for an interview, with selection seeking to maximize diversity in gender (50% male), race (12.5% non-White), and drug of choice (opiates, methamphetamine, and cocaine; Table 3).
Methods are also a little underdone for qualitative, who did the interviews (were they independent of the clinical trial), who did the analysis, was coding independently triangulated? Suggest using COREQ checklist to ensure data trustworthiness can be demonstrated.
- A description of who performed the interviews (and surveys) is now included in METHODS. As with the limitations associated with small sample size, more thorough and rigorous qualitative analyses and checks beyond that already described were not performed.
I do not agree with findings being presented as bullet points and should be written in regular paragraph styles.
- Generally speaking, we agree and have changed the cancer screening results to subheadings. Still, for the SHE questions, we have kept as bullet points due to the fact that they are each a single sentence and not readily related to each other (lesser flow continuity in a paragraph). Still, we can revise to a paragraph if needed.
Line 29 – what is SHE? Is this referring to survey items?
- The following statement has been added to METHODS: We also asked four questions specifically related to healthcare access and use in relation to their drug use. Termed stigmatizing healthcare experiences (SHE), they were as follows: SHE-1: Have you ever avoided health care because of your drug use?; SHE-2: Have you ever had a healthcare worker refuse to treat you or denied you access to medical treatment or care because of your drug use?; SHE-3: Do you feel 'accepted' and 'non-judged' during office visits?; and SHE-4: Have you ever had bad experiences with primary care such that you considered not going anymore?
Be consistent with excerpts – all in italics
- We have made them consistent.
Reviewer 3 Report
This is novel study conducted in a hard to study population. The small sample size is understandable with recruitment challenges and described in the limitations.
Author Response
No response needed/no comments to address.